# Oral Complications of ICU Patients with COVID-19: Case-Series and Review of Two Hundred Ten Cases

**DOI:** 10.3390/jcm10040581

**Published:** 2021-02-04

**Authors:** Barbora Hocková, Abanoub Riad, Jozef Valky, Zuzana Šulajová, Adam Stebel, Rastislav Slávik, Zuzana Bečková, Andrea Pokorná, Jitka Klugarová, Miloslav Klugar

**Affiliations:** 1Department of Maxillofacial Surgery, F. D. Roosevelt University Hospital, 975 17 Banska Bystrica, Slovakia; bhockova@nspbb.sk (B.H.); astebel@nspbb.sk (A.S.); rslavik@nspbb.sk (R.S.); 2Department of Prosthetic Dentistry, Faculty of Medicine and Dentistry, Palacky University, 775 15 Olomouc, Czech Republic; 3Czech National Centre for Evidence-Based Healthcare and Knowledge Translation (Cochrane Czech Republic, Czech EBHC: JBI Centre of Excellence, Masaryk University GRADE Centre), Institute of Biostatistics and Analyses, Faculty of Medicine, Masaryk University, 625 00 Brno, Czech Republic; apokorna@med.muni.cz (A.P.); klugarova@med.muni.cz (J.K.); klugar@med.muni.cz (M.K.); 4Department of Public Health, Faculty of Medicine, Masaryk University, 625 00 Brno, Czech Republic; 5Department of Anaesthesiology, F. D. Roosevelt University Hospital, 975 17 Banska Bystrica, Slovakia; jvalky@nspbb.sk (J.V.); zsulajova@nspbb.sk (Z.Š.); 6Department of Clinical Microbiology, F. D. Roosevelt University Hospital, 975 17 Banska Bystrica, Slovakia; zbeckova@nspbb.sk; 7St. Elizabeth University of Health and Social Work, 812 50 Bratislava, Slovakia; 8Department of Nursing and Midwifery, Faculty of Medicine, Masaryk University, 625 00 Brno, Czech Republic

**Keywords:** candidiasis, COVID-19, critical care, macroglossia, oral manifestations, pressure ulcer, prone position

## Abstract

Background: The critically ill patients suffering from coronavirus disease (COVID-19) and admitted to the intensive care units (ICUs) are susceptible to a wide array of complications that can be life-threatening or impose them to long-term complications. The COVID-19 oral mucocutaneous complications require multidisciplinary management and research for their pathophysiological course and epidemiological significance; therefore, the objective of this study was to evaluate the prevalence and characteristics of the critically ill COVID-19 patients with oral complications. Methods: We described the clinical and microbiological characteristics of the critically ill COVID-19 patients in our ICU department (Banska Bystrica, Slovakia). In addition, we reviewed the current body of evidence in Ovid MEDLINE^®^, Embase, Cochrane Library, and Google Scholar for the oral mucocutaneous complications of ICU patients with COVID-19. Results: Three out of nine critically ill patients (33.3%) in our ICU department presented with oral complications including haemorrhagic ulcers and necrotic ulcers affecting the lips and tongue. The microbiological assessment revealed the presence of opportunistic pathogens, confirming the possibility of co-infection. On reviewing the current literature, two hundred ten critically ill patients were reported to have oral complications due to their stay in the ICU setting. Perioral pressure ulcers were the most common complication, followed by oral candidiasis, herpetic and haemorrhagic ulcers, and acute onset macroglossia. The prolonged prone positioning and mechanical ventilation devices were the primary risk factors for those oral complications, in addition to the immunosuppressive drugs. Conclusions: The multidisciplinary approach is strongly advocated for monitoring and management of COVID-19 patients, thus implying that dermatology and oral healthcare specialists and nurses should be integrated within the ICU teams.

## 1. Introduction

The patients with coronavirus disease (COVID-19) have been diagnosed with an array of oral and dermatologic symptoms in addition to their typical respiratory manifestations [1,2,3,4,5,6,7]. These oral symptoms were equally distributed across the gender and had higher prevalence among older patients and the patients with higher severity of the COVID-19 infection [1,2]. However, there is still a question about the pathophysiologic origin of these symptoms, whether they are due to direct viral infection, co-infections, drug reactions, iatrogenic complications, or stress [2]. Current research shows that coronavirus damage to respiratory and other organs could be related to the distribution of angiotensin-converting enzyme 2 (ACE-2) receptors in the human body [8]. The high expression of ACE-2 in the epithelial cells of the tongue and the salivary glands may explain the development of dysgeusia and the mucocutaneous oral lesions in patients with COVID-19 [9]. 

The epidemiologic evidence reveals that up to one-quarter of the hospitalised COVID-19 patients need intensive care unit (ICU) admission, making them more vulnerable to secondary pneumonia, cardiac injury, sepsis, kidney injury, and neurologic disorders [10]. The ICU-related cutaneous and mucosal complications, including contact dermatitis, cutaneous candidiasis, pressure ulcers, and hospital-acquired infections (HAIs) have been well documented, and they require a multidisciplinary approach for timely diagnosis and treatment [11].

The association of oral health and critical care can be depicted as a bidirectional relationship because frequent toothbrushing and the use of chlorhexidine were found by a recent Cochrane review to be effective in preventing ventilator-associated pneumonia in critically ill patients [12]. Moreover, in a national retrospective analysis of ICU patients in the Czech Republic, facial pressure ulcers including perioral ulcers were significantly associated with the length of stay in the ICU [13].

The severity and frequency of dermatologic disorders increase dramatically in the patients with prolonged ICU stay; therefore, a tight collaboration among intensivists, anesthesiologists, dermatologists, nurses, and oral healthcare professionals cannot be emphasised enough especially for critically ill COVID-19 patients [14].

In compliance with this guideline, we performed an oral examination for all the patients at our ICU department with COVID-19 in order to evaluate the hypothesis of critical care impact on oral mucocutaneous conditions emergence.

The primary objectives of this study were to evaluate the prevalence of oral complications in ICU patients with COVID-19 and to describe these oral mucocutaneous conditions clinically and microbiologically if present. The secondary objective was to review the current body of evidence regarding the oral mucocutaneous complications of ICU patients with COVID-19.

## 2. Materials and Methods

On 8 December 2020, there were nine critically ill COVID-19 patients at the ICU department of F.D. Roosevelt Teaching Hospital (Banska Bystrica, Slovakia). All the patients underwent a complete oral examination by the same investigator who also took nasopharyngeal and lingual swabs for microbiological assessment and reverse transcription-polymerase chain reaction (RT-PCR) testing.

The samples were collected under sterile conditions in the morning on an empty stomach using a cotton swab with a solid transport medium for microbiological assessment. On standard culture media prepared by MASTERCLAVE 10^®^ (Biomérieux, Marcy-l’Étoile, France), the upper respiratory tract’s biological materials were cultivated using Columbia agar base dehydrated with sheep blood (Columbia 64674 Bio-Rad, Marnes-la-Coquette, France), chocolate agar base dehydrated with horse blood (Columbia 64678 Bio-Rad, Marnes-la-Coquette, France), and Sabouraud dextrose agar base dehydrated (Sabouraud 64494 Bio-Rad, Marnes-la-Coquette, France) in a biological thermostat for 18–24 h at a temperature of 35–37 °C. In the case of pathogenic microorganisms, sensitivity to antibiotics was determined qualitatively by the disk diffusion method and quantitatively by estimating minimum inhibitory concentration (MIC) using the modified microdilution method. 

For RT-PCR testing, a viral transport medium (VTM) compatible with the isolation kit, inactivating the accompanying microbial flora and stabilising the nucleic acid, was used. The samples were collected in the morning on an empty stomach, and the patients coughed before swabbing. The collection set included two pieces of Dacron collection tampons. Firstly, the investigator wiped the palatal arches with a tampon in a circular motion without touching the tonsils. Secondly, the investigator wiped the mucosa of the nasal dome back through both nostrils. The samples were transported within one hour to the microbiology laboratory of the hospital to be tested using an RT-PCR system with a thermal cycler and QuantStudio^TM^ 5 (Thermo Fisher Scientific Inc., Waltham, MA, USA). VERSANT^®^ Sample Preparation 1.0 Reagents Kit (Siemens AG. Munich, Germany) was used to isolate the viral RNA, which was transcribed into complementary DNA (cDNA) and amplified by standard polymerase chain reaction methods. For viral detection, the FTD SARS-CoV-2 Assay (Siemens AG. Munich, Germany) kit which identifies N and ORF1ab genes was used.

The oral cavity was systematically examined—beginning with palatoglossal arch, followed by mucosa of palatum durum et molle, upper and lower gingiva, dorsum of the tongue, buccal mucosa, and floor of the mouth. In intubated patients, the examination was more challenging to perform. The patients’ medical and oral anamneses had been reported according to the CARE guidelines and in full accordance with the Declaration of Helsinki for medical research involving human subjects [15,16]. 

In the second part of this study, we searched the literature from inception until 30 December 2020 for ICU-related oral conditions in COVID-19 patients. An electronic search strategy composed of a combination of keywords ((COVID-19 OR SARS-CoV-2 OR coronavirus) AND (candidiasis OR ulcer OR macroglossia OR xerostomia)) was developed and carried out in Ovid MEDLINE^®^, Embase, Cochrane Library, and Google Scholar. The inclusion criteria were admission to ICU and COVID-19 confirmation by RT-PCR testing. The outcomes of interest included all oral mucocutaneous conditions regardless of their severity and duration. No restrictions for language or study type were applied.

## 3. Results

In our examined series of cases, various oral conditions were found in three (33.3%) of them. The most common condition was haemorrhagic ulceration. On microbiological assessment, *Pseudomonas aeruginosa* was cultivated in two (22.2%) patients, and *Klebsiella pneumoniae* and *Enterococcus faecalis* were cultivated in one (11.1%) patient. The RT-PCR testing yielded positive results in both nasopharyngeal swabs and lingual swabs of 66.6% of the patients with oral conditions who are further described in detail (Table 1).

### 3.1. Case-Series of ICU Patients in Banska Bystrica

#### 3.1.1. Case Report No. 1

A 68-year-old male patient with arterial hypertension, chronic hepatopathy, hypercholesterolemia, and gastroesophageal reflux disease tested positive during the mass antigen-based testing in Slovakia on 31 October 2020 [17]. Three days later, he progressed with headache, fever, dry cough, and dyspnoea. He was transferred on 8 November 2020 from his district hospital in west Slovakia to our ICU department due to an occupancy issue. At our ICU department, the patient was continuously under analgosedation by Tramadol and *Tiapridal* (Tiapride; Sanofi-Aventis, Bratislava, Slovakia), and he began to receive *Entizol* (Metronidazole; Polpharma, Warsaw, Poland) because of clostridium difficile from 12 November 2020. Two days later, he began to receive Cefepime and Colistin due to *Pseudomonas aeruginosa* which were replaced by Vancomycin and Meropenem on 16 November 2020. To control hypercoagulation, the patient received *Fraxiparine* (Nadroparin calcium; GlaxoSmithKline Slovakia, Bratislava, Slovakia) which was replaced by Heparin, and to control clostridia infection, the patient was prescribed Noradrenalin, Dobutamine, and *Embesin* (Vasopressin; Orpha-Devel Handels und Vertriebs GmbH, Purkersdorf, Austria). By 29 November 2020, the clinical condition worsened, and the inflammatory parameters increased; therefore, Piperacillin/tazobactam, Linezolid, and Voriconazole were administered.

On 8 December 2020, the patient was intubated, under sedation and without tracheostomy. Our oral examination found out oral lesions at the dorsal surface of the tongue, specifically in the middle third, in the form of haemorrhagic ulcerations. Oral mucosa of the mouth in other places was free of lesions. The microbiological assessment for the swab of tongue dorsum showed *Pseudomonas aeruginosa,* and the RT-PCR testing for severe acute respiratory syndrome coronavirus 2 (SARS-CoV-2) was positive for both nasopharyngeal and lingual swabs. During the evening check-up, oedema of the masseter region was observed bilaterally with pain on palpation which was examined by the head of the maxillofacial surgery department who confirmed an acute form of bilateral parotitis. Non-invasive treatment was carried out using a sterile swab from the exudate for cultivation then administering antibiotic therapy.

The patient deceased on 11 December 2020 due to septic shock and multiple organ dysfunction syndrome induced by enterocolitis. According to the Committee on Publication Ethics’ (COPE) Code of Conduct, the permission of the patient’s next of kin was granted [18].

#### 3.1.2. Case Report No. 2

A 61-year-old male polymorbid obese patient with arterial hypertension and a history of myocardial infarction and septic shock was admitted to the ICU department with bilateral COVID-19-related pneumonia on 12 November 2020. After two weeks, tracheostomy was done by an otolaryngologist. On 2 December 2020, a dermatologist was consulted due to exanthema on the skin of shoulders and back which was diagnosed as viral exanthema commonly observed in COVID-19 patients [19]. A topical treatment protocol was composed of *bisulepin* (Dithiaden; Zentiva, Prague, Czech Republic), *loratadine* (Flonidan; TEVA Pharmaceuticals Slovakia, Bratislava, Slovakia), and *betamethasone* (Beloderm; Fagron, Olomouc, Czech Republic).

During the oral examination, the patient was not under sedation. The microbiological assessment confirmed the presence of *Enterococcus faecalis* and *Pseudomonas aeruginosa*. However, his RT-PCR testing for lower respiratory tract sputum yielded a positive result on December 7th and 14th, the results of nasopharyngeal and lingual swabs negative. The intraoral examination revealed multiple lesions located on the tongue dorsum and labial mucosa. The lesions were mainly haemorrhagic ulcerations along the lips and focal necrosis affecting the anterior third of tongue dorsum accompanied by white patches. On December 18th, tracheostomy cannula was removed, and three days later, the patient tested negative and was transferred to the non-COVID-19 department.

#### 3.1.3. Case Report No. 3

A 64-year-old male patient who had had contact with his COVID-19 positive daughter was examined at the emergency department of our hospital with moderate symptoms of COVID-19; therefore, he was treated at home due to persistent fever, dyspnoea, and dry cough. The general practitioner prescribed him an antibiotic treatment of cefixime one week before performing an antigen test for SARS-CoV-2, which yielded a positive result. On 14 November 2020, the patient was admitted to our ICU department, where a computed tomography (CT) scan revealed severe bronchopneumonia requiring oxygen supplement. During his ICU stay, the patient was intubated and treated with Methylprednisolone and Remdesivir.

On 8 December 2020, the patient was not under sedation; therefore, he was able to communicate nonverbally. The extraoral examination showed viral exanthem in the form of painless macules on the skin, while the intraoral examination yielded focal painful lesions located mainly along the upper and lower lip with a maximum diameter of 7 mm and erythema around them thus resembling haemorrhagic ulcerations. The oral lesions developed simultaneously with ICU admission. On 14th and 16th December 2020, the patient tested negative; therefore, he was transferred to the non-COVID-19 department.

### 3.2. Literature Review

On reviewing the emerging evidence on ICU-related complications in COVID-19 patients, fourteen studies (one cohort study [20], two case-control studies [21,22], one cross-sectional study [23], two case-series [24,25], and eight case-reports [26,27,28,29,30,31,32,33]) with two hundred ten patients met the inclusion criteria. The majority of the cases were from the Americas (USA *n* = 103, 49.5%; Brazil *n* = 4, 1.9%), followed by Europe (Spain *n* = 57, 27.1%; UK *n* = 16, 7.6%; France *n* = 2, 1%; and Italy *n* = 2, 1%), and Middle East (Iran *n* = 26, 12.4%). The demographic characteristics of 85 cases were described in the primary studies, sixty-two of them (72.9%) were males, and twenty-three (27.1%) were females. The reported cases’ average age was 60.2 years old (min: 27, max: 81 years old). All the patients had been defined as critically ill according to the Australian guidelines for the clinical care of people with COVID-19 [34]. On their hospital admission, the patients were initially treated with antibiotics, corticosteroids, and hydroxychloroquine sulphate just before transferring to the ICU department. During their ICU stay, one hundred eighty patients (85.7%) underwent prone positioning in addition to mechanical ventilation (Table 2).

The reported oral complications were mainly perioral pressure ulcers (*n* = 179, 85.2%), intraoral candidiasis (*n* = 27, 12.9%), other intraoral ulcers (*n* = 3, 1.4%), and macroglossia (*n* = 1, 0.5%). While the onset ranged between four and twenty-four days after ICU admission, the duration ranged between one and two weeks. The medical treatments included dressings, position adjustment, antifungals, antivirals, and surgical interventions including full-thickness excisions. Regarding the suggested aetiology, most of the complications were caused by the prone positioning which is an essential procedure for some cases in critical care. Prolonged pronation cycles, pronation monitored by less experienced staff, and use of respiratory support equipment were risk factors to increase the incidence of pressure ulcers among ICU patients who underwent pronation. The use of broad-spectrum antibiotics and immunosuppressive drugs was associated with co-infections such as fungal, bacterial, or viral infections in the hospital setting (Figure 1).

#### 3.2.1. Perioral Pressure Ulcers (ICD-11: EH90)

The perioral (facial) pressure ulcers have been the most prevalent ICU-related oral complication in COVID-19 patients reported by ten studies in one hundred seventy-nine patients; 73.75% of them were males, and the vast majority were of old age [20,21,22,23,25,29,30,31,32,33]. Prolonged pronation and endotracheal intubation were the most evident risk factors for perioral pressure ulcers. Given the long-term psychological impact of scarring caused by perioral ulcers, and their interference with mechanical ventilation equipment in the critical care setting, an array of interventions and prophylactic precautions has been proposed to prevent this potential epidemic [35,36,37].

#### 3.2.2. Oral Candidiasis (ICD-11: 1F23.0)

Oral (oropharyngeal) candidiasis has been reported by two studies with twenty-seven patients who have been treated initially by broad-spectrum antibiotics and immunosuppressants, which are believed to cause immune dysregulation [24,26]. Therefore, the reported patients were more susceptible to get secondary infections and HAIs, and they were managed by either systemic fluconazole or topical nystatin according to the infection severity and lesion surface. In addition to our included cases, there were thirty-six COVID-19 patients with milder clinical courses who experienced oral candidiasis and were not admitted to the ICU [24,38,39,40,41,42,43]. The most common risk factor among mild, moderate, and critically ill COVID-19 patients with fungal co-infection, e.g., oral candidiasis, was the prolonged use of antibiotics.

#### 3.2.3. Oral Ulcers (ICD-11: DA01)

Herpetic ulcers and haemorrhagic necrotic ulcers were reported by Brandao et al. 2020 in three patients above 70 years old [28]. The ulcers emerged four-to-five days after the respiratory symptoms, and they were treated by antivirals and photobiomodulation therapy. The lesions were suggested to be triggered by the ICU admission and the pre-admission antibiotics, which may have caused immune dysregulation, thus promoting HAIs.

#### 3.2.4. Macroglossia (ICD-11: DA03.5)

Acute macroglossia has been reported by Andrews et al. 2020 in a 40-year-old male patient who experienced prolonged pronation cycles for eleven days [27]. Endotracheal tubes and throat packing as well had been associated with lingual oedema as a result of disruption of venous drainage [44]. There has been a number of ICU cases with acute macroglossia before the COVID-19 pandemic; therefore, the emergence of this rare but devastating complication was anticipated, and its non-invasive management is deemed required [27].

## 4. Discussion

The emerging evidence on COVID-19 related oral manifestations had triggered a broad debate regarding the pathophysiological course and the epidemiological significance of these mucocutaneous symptoms, given that the case definition of COVID-19 needs to be as sensitive as possible [1,2,3,4,5,6,7,43]. The current case definitions of COVID-19 are exclusively dependent on the typical pulmonary symptoms common with other respiratory diseases. Meanwhile, from the laboratory perspective, leukopenia with lymphopenia, thrombocytopenia, high values of C-reactive proteins, and low levels of procalcitonin are well-established diagnostic aids for case triage [45]. In the review of Iranmanesh et al. 2020, direct viral enanthem, inflammatory response secondary to the viral infection, opportunistic infections, lack of oral hygiene, and stress were suggested aetiologies for the oral symptoms in COVID-19 patients which were equally distributed across the gender and associated with older age and more severe clinical courses of the disease [1]. As highlighted by Riad et al. 2020, the lack of reference time point consistency among the COVID-19 case-reports and case-series has undermined the efforts for accurate estimation of the onset of the COVID-19 related oral symptoms and their epidemiological significance [2]. Therefore, following the reporting guidelines was and is still strongly advocated for COVID-19 clinical literature.

The COVID-19 critically ill population imposed an unprecedented challenge for the health systems worldwide due to the supply/demand ratio which has been further complicated by lack of evidence on the clinical prognosis of COVID-19 cases and their post-admission complications including the life-threatening ones [46]. However, the common terminating complications are related to acute respiratory distress syndrome (ARDS) like multi-organ failure, kidney injury, sepsis, atrial arrhythmias, and myocardial infarction, the dermatological complications like candidiasis, and perioral ulcers can become life-threatening if left untreated [10]. Therefore, the multidisciplinary approach of managing COVID-19 patients in ICU units, the use of teledermatology and teledentistry, the allied staff’s awareness of those mucocutaneous complications are highly recommended while navigating through this pandemic [47,48,49].

The primary objective of this study was to share our clinical experience with critically ill COVID-19 patients in a central European country with 120,203 detected cases (63,818 females and 56,385 males), 1046 deceased cases, and 30,753 active cases on the day of our clinical examination—8 December 2020 [50]. The oral complications of our ICU patients were similar to those described by Brandao et al. 2020 from Brazil, as both groups of patients experienced haemorrhagic ulcers related to the lips and labial mucosa [28]. All the patients were above 60 years old, and they were mainly males with a pre-ICU antibiotic therapeutic course which had been extended during their ICU stay; therefore, immune dysregulation was suggested as a pathophysiological pathway. This suggestion was supported by the microbiological results of our patients which revealed the increase of opportunistic species, e.g., *Pseudomonas aeruginosa*, *Klebsiella pneumoniae*, and *Enterococcus faecalis*.

One of the limitations of this case-series is that the microbiological assessment of the lingual swabs was carried out for the patients with oral complications only, and it was not possible to take clinical photographs because of infection control guidelines and the fact that there was only one clinical investigator permitted to examine all the patients.

Provided that prone positioning-related complications were the most prevalent ICU-related oral complications, Moore et al. 2020 systematically reviewed the current body of evidence and recommended to use pressure redistribution support surface/positioning devices, use protective coverings during pronation, and carry out simple and frequent changes in the posture of the patient and the device positioning. The clinicians are strongly advised to assess the common risk areas for pressure ulcers frequently and to keep the skin clean and moisturized [35]. This practice recommendation requires a close collaboration between intensivists, anaesthesiologists, nurses, dermatologists, and dentists to monitor and manage the pressure ulcers at an early stage. Overloaded healthcare systems for the long-term by COVID-19 might be one of the reasons for the high prevalence of perioral pressure ulcers as in many countries less experienced healthcare staff might be present in the ICU as retrieved by our literature review [22,29]. 

The immune dysregulation was suggested as a pathophysiological pathway for the emergence of oral ulcers in COVID-19 patients, especially in the severely affected ones. This hypothesis was supported by several cases where recurrent aphthous stomatitis and traumatic ulcerations were ruled out based on rigorous clinical and laboratory investigation, while herpes simplex virus was detected in the vast majority of the old and immunocompromised patients [28,51]. Reflecting on other immune dysregulation-related oral complications like the opportunistic infections especially those that emerge in the hospital setting, the review of Rawson et al. 2020 recommended to develop antimicrobial stewardship protocols for managing COVID-19 patients in order to support the optimal treatment outcomes and prevent the potential bacterial/fungal co-infection in critically ill COVID-19 patients [52].

Although it is currently confirmed by several high-quality clinical practice guidelines that hydroxychloroquine should not be used as the treatment for COVID-19 [34,53,54]. This research waste was identified in treatment protocols of cases from Brazil, Italy, and the USA. Higher awareness and evidence-based medicine principles about COVID-19 treatment should be advocated in all countries. A very useful tool that can inform practice by best available evidence might be the recently published living COVID-19 recommendation map [55,56]. 

## 5. Conclusions

Perioral pressure ulcers, oral candidiasis, herpetic and haemorrhagic oral ulcers, and acute macroglossia were the commonly reported complications in critically ill COVID-19 patients. These oral mucocutaneous complications were caused by the prolonged prone positioning and mechanical ventilation devices in the ICU setting, in addition to the immunosuppressive treatments prescribed for this special cohort of patients. Therefore, a multidisciplinary approach is strongly advocated for monitoring and management of COVID-19 thus implying that dermatology and oral healthcare specialists and nurses should be integrated within the ICU teams.

## Figures and Tables

**Figure 1 jcm-10-00581-f001:**
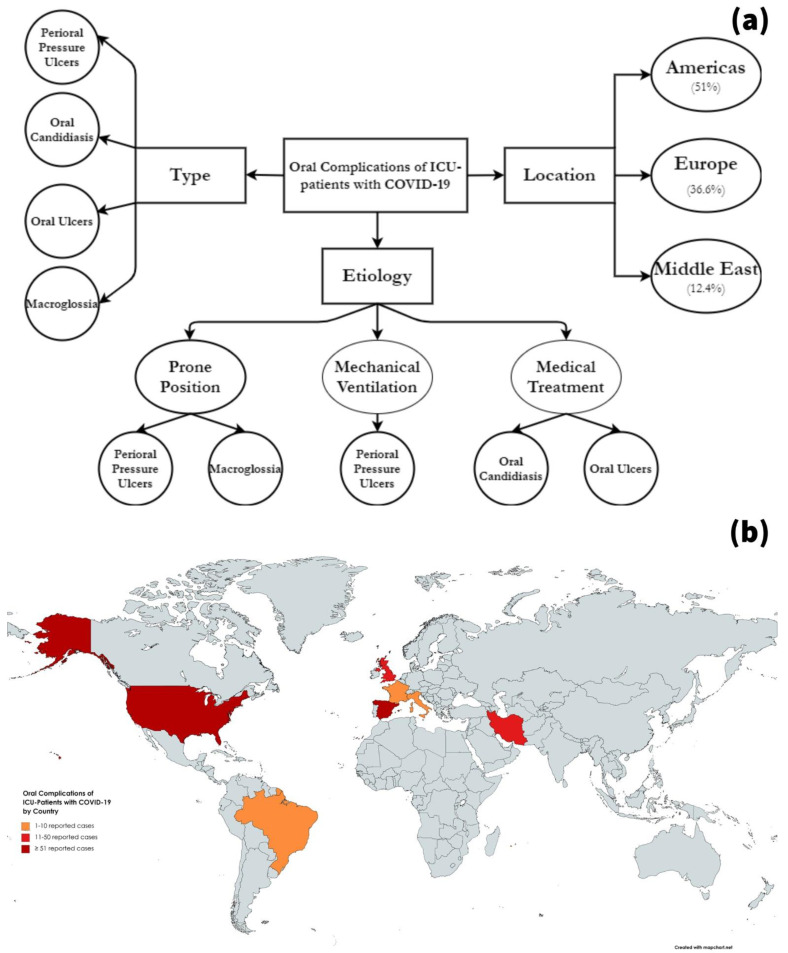
Summary of the oral complications which were reported in the ICU patients with COVID-19; (**a**) by type, aetiology, and region; (**b**) frequency of reported cases by country.

**Table 1 jcm-10-00581-t001:** Critically ill COVID-19 patients at F.D. Roosevelt Teaching Hospital (SK)—8 December 2020.

	Patient No. 1	Patient No. 2	Patient No. 3
Age, gender	68-year-old, Male	61-year-old, Male	64-year-old, Male
Medical anamnesis	Arterial hypertension, chronic hepatopathy, hypercholesterolemia, and gastroesophageal reflux disease.	Obesity, arterial hypertension, and a history of myocardial infarction and septic shock. **Chronic medications**: Egiramlon (Ramipril), Ebrantil (Urapidil), Tenaxum (Rilmenidine), and Metoprolol.	**Chronic medications**: Coaxil (Tianeptine), Trittico (Trazodone), and Cefixime.
COVID-19-related treatment before ICU admission	Ceftriaxone, Klacid (Clarithromycin), Remdesivir, Paracetamol, Solumedrol (Methylprednisolone), Vitamin C, Vitamin B1, and Fraxiparine (Nadroparin calcium). **Nasogastric tube**: Isoprinosine (Inosine pranobex), Atorvastatin, Lagosa, Vigantol (Cholecalciferol), Zinc, and Quamatel (Famotidine).	Ceftriaxone, Remdesivir, Dexamethasone, Polyoxidonium, Vitamin C, Vitamin B1, and Fraxiparine (Nadroparin calcium). **Nasogastric tube**: Isoprinosine (Inosine pranobex), Atorvastatin, Lagosa, Vigantol (Cholecalciferol), Zinc, and Quamatel (Famotidine).	Cefixime, Remdesivir, Solumedrol (Methylprednisolone).
Date of ICU admission	8 November 2020	12 November 2020	14 November 2020
Dermatologic complications (extraoral)	No	Yes	Yes
Oral examination	Haemorrhagic ulcers in the middle third of the dorsal surface of the tongue.	Haemorrhagic ulcers along the lips and focal necrosis affecting the anterior third of the dorsal surface of the tongue accompanied by white patches.	Painful haemorrhagic ulcers along the upper and lower lips. Viral exanthem on the skin in the form of painless macules.
Reverse transcription-polymerase chain reaction (RT-PCR) ^1^	LRTS: positive Nasopharyngeal: positive Lingual: positive	LRTS: positive Nasopharyngeal: negative Lingual: negative	LRTS: positive Nasopharyngeal: positive Lingual: positive
Microbiological assessment	*Pseudomonas aeruginosa*	Gram-positive cocci, *Enterococcus faecalis,* and *Pseudomonas aeruginosa*	*Klebsiella pneumoniae*
Post-ICU outcomes	Deceased on 11 December 2020 due to septic shock and multiple organ dysfunction syndrome.	Released from ICU on 21 December 2020 in good condition and without intubation.	Released from ICU on 17 December 2020 after two consecutive negative RT-PCR results on December 14th and 16th, 2020.

^1^ LRTS: lower respiratory tract sputum.

**Table 2 jcm-10-00581-t002:** Summary of the oral complications of ICU patients with COVID-19.

Study, Location	*N*	Gender	Age	MED-COVID-19	Complication	Onset	Duration	MED-Oral	Aetiology
Amorim dos Santos et al. [26] 2020; Brasilia (Brazil)	1	Male	67	Initially: hydroxychloroquine sulphate, ceftriaxone sodium, and azithromycin. Later: meropenem, sulfamethoxazole, trimethoprim, immunosuppressants, and anticoagulants.	Oral Candidiasis	After 24 days of ICU admission.	2 weeks.	Systemic fluconazole and oral nystatin.	Opportunistic fungal infection due to COVID-19 treatments (antibiotics) which lead to immune dysregulation.
Andrews et al. [27] 2020; (Michigan, USA)	1	Female	40	Initially: hydroxychloroquine sulphate, methylprednisolone, and tocilizumab. Later: 16 h. prone/8 h. supine (11 days).	Acute Macroglossia	After 11 days of prone position.	11 days.	Lingual compression.	Prolonged course of prone positioning for treatment of COVID-19.
Brandao et al. [28] 2020; (Sao Paolo, Brazil)	3	1 Female 2 Male	81, 71, and 72	Azithromycin, piperacillin/tazobactam, and ceftriaxone.	Herpetic Ulcers and Haemorrhagic Ulcers	After 5, 4, and 5 days of respiratory symptoms.	11, >15, and 7 days.	Acyclovir and PBMT.	The ICU admission leads to immune dysfunction.
Ibarra et al. [22] 2020; (Madrid, Spain)	57	16 Female 41 Male	μ = 61 years old (56–69)	16 h. prone/8 h. supine (μ = 6 days).	Perioral Pressure Ulcers	N/A	N/A	All the cases were managed with dressings.	Prone position pressure ulcers are dependent on the clinical manoeuvre.
Martel et al. [21] 2020; (Massachusetts, USA)	18	N/A	N/A	N/A	Perioral Pressure Ulcers	N/A	N/A	All the cases were managed with dressings.	Prolonged placement in a prone position and use of respiratory support equipment.
Perrillat et al. [29] 2020; (Marseille, France)	2	2 Male	27, 50	6 prone sessions (12 h. each) 9 prone sessions (12 h. each)	Perioral Pressure Ulcers	N/A	N/A	Debridement of necrotic tissue and paraffin gauze dressing.	Prone positioning monitored by less experienced staff.
Ramondetta et al. [30] 2020; (Turin, Italy)	1	Male	48	Initially: hydroxychloroquine and antiviral treatment. Later: prone positioning cycles (5 days).	Perioral Pressure Ulcers	After 15 days of ICU admission.	N/A	Advanced dressings.	The pressure exercised during the prone positioning phases.
Rekhtman et al. [20] 2020 (New York, USA)	13	N/A	N/A	Mechanical ventilation.	Perioral Pressure Ulcers	N/A	N/A	N/A	Increased pressure from endotracheal tubes or medical equipment used to hold tubes in place.
Salehi et al. [24] 2020; (Tehran, Iran)	26	N/A	N/A	Antiviral, antibacterial, and corticosteroids.	Oropharyngeal Candidiasis	N/A	N/A	Fluconazole, nystatin and caspofungin.	Immune dysregulation due to ICU admission and broad-spectrum antibiotic.
Shearer et al. [23] 2020; (Washington DC, USA)	68	N/A	N/A	Mechanical ventilation. μ of prone positioning = 5.14 days (1–26).	Perioral Pressure Ulcers	N/A	N/A	Antimicrobial dressings.	Longer duration of prone position appears to confer greater risk for developing these pressure injuries.
Singh et al. [31] 2020; (California, USA)	2	1 Female 1 Male	44, 71	Directly admitted to the ICU, a course of azithromycin and prednisone at home. Mechanical ventilation.	Perioral Pressure Ulcers	4 days after prone positioning; 5 days after prone positioning.	N/A	Pressure-relieving turns and position system, bordered foam dressings, fluidised positioners.	Prone positioning.
Siotos et al. [32] 2020; (Illinois, USA)	1	Female	82	Mechanical ventilation.	Perioral Pressure Ulcers	10 days after prone positioning.	N/A	Surgical removal by full-thickness excision.	Prone positioning.
Sleiwah et al. [25] 2020; (London, UK)	16	2 Female 14 Male	μ = 58.6 years old (40–77)	Mechanical ventilation, and vasopressors. μ of prone positioning = 5.2 days (2–7).	Perioral Pressure Ulcers	N/A	N/A	N/A	Prone positioning.
Zingarelli et al. [33] 2020; (Alessandria, Italy)	1	Female	50	Initially: acetaminophen and antibiotics. Later: ICU admission, mechanical ventilation.	Perioral Pressure Ulcers	After 15 days of ICU admission.	1 week	Advanced dressings.	Prone positioning.
Total	210	23 Female 62 Male	μ = 60.2 years old (27–81)	Antibiotics, hydroxychloroquine, corticosteroids, mechanical ventilation, and prone positioning.	Perioral pressure ulcers, candidiasis, stomatitis, and macroglossia.	Range 4–24 days after ICU admission.	Range 1–2 weeks.	Dressings, position change, antifungals, and antivirals.	Prone positioning, mechanical ventilation, antibiotic therapy, HAIs.

N: number of patients; MED-COVID-19: medical treatment for COVID-19 and its complications; MED-Oral: medical treatment for the oral complications; N/A: not reported by the authors; PBMT: photobiomodulation therapy.

## Data Availability

The data that support the findings of this study are available from the corresponding author upon reasonable request.

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
