# Peer review of "Oral Complications of ICU Patients with COVID-19: Case-Series and Review of Two Hundred Ten Cases"

_jcm, 2021, doi:10.3390/jcm10040581_

Round 1
Reviewer 1 Report
Authors presented good case article combined with review. It's a pity they describe only 9 own cases, but it is related to the clinic. I suggest write some more about general clinical symptoms of SARS-2 infection. Please see to the following article and cite it: https://sites.kowsarpub.com/jjm/articles/103744.html In the article should be correct name of PCR, because for detection of SARS-2 RNA is used reverse transcriptase real-time PCR, not normal PCR. Moreover, in Materials and Methods should be described in details real-time PCR and microbiological assessment, presented also in Table 1. It would be great if the Authors would add photos of some oral manifestations found in patients. What antibiotics were used in bacterial infections treatment? Literature review is very good presented. In Table 2 should be changed "No." into "No. of patients". Figure 1A has poor quality, also the font in the circles should be larger.
Author Response
Dear Reviewer,
On behalf of my fellow co-authors, I would like to thank you for your esteemed effort and time for reviewing our manuscript “Oral Complications of ICU Patients with COVID-19: Case-Series and Review of Two Hundred Ten Cases” submitted to Journal of Clinical Medicine (JCM).
In response to your suggestions, we have amended our manuscript accordingly:
- “In the article should be correct name of PCR, because for detection of SARS-2 RNA is used reverse transcriptase real-time PCR, not normal PCR.”
The name has been corrected to be reverse transcription-polymerase chain reaction (RT-PCR). Refer to Line No. 89.
- “I suggest write some more about general clinical symptoms of SARS-2 infection. Please see to the following article and cite it: https://sites.kowsarpub.com/jjm/articles/103744.html”
We added a sentence in the discussion regarding the case definitions of COVID-19, and the suggested article had cited. Refer to Line No. 273-276.
- “Moreover, in Materials and Methods should be described in details real-time PCR and microbiological assessment, presented also in Table 1.”
We added a detailed description for microbiological assessment and RT-PCR testing method. Refer to Line No. 90-109.
- “It would be great if the Authors would add photos of some oral manifestations found in patients.”
We have explained in the limitations paragraph that it was not possible to take clinical photographs because of infection control guidelines and the fact that there was only one clinical investigator allowed to examine all the patients. Refer to Line No. 310-313.
- “In Table 2 should be changed "No." into "No. of patients".”
Corrected. N: number of patients. Refer to Table 2.
- “Figure 1A has poor quality, also the font in the circles should be larger.”
The figure has been fixed and the font has been enlarged. Refer to Figure 1.A.
Sincerely,

Reviewer 2 Report
Introduction provides sufficient details and it is focused on introducing the background of the aim of the paper. Please consider citing a recently published article on MDPI that evaluates the use of mobile apps and telemedicine to assist patients and operators (doi: 10.3390/jcm9061891). It could help to broaden your introduction
Methods are well-organized and accurately described
Results appear clearly presented and undestandable
Discussion and conclusion are supported by results and provide adequate context with the existing literature
Author Response
Dear Reviewer,
On behalf of my fellow co-authors, I would like to thank you for your esteemed effort and time for reviewing our manuscript “Oral Complications of ICU Patients with COVID-19: Case-Series and Review of Two Hundred Ten Cases” submitted to Journal of Clinical Medicine (JCM).
In response to your suggestions, we have amended our manuscript accordingly:
- “Please consider citing a recently published article on MDPI that evaluates the use of mobile apps and telemedicine to assist patients and operators (doi: 10.3390/jcm9061891).”
We have amended a paragraph in the discussion part regarding the use of teledentistry during this pandemic. The suggested article has been cited. Please Refer to Line No. 294-297, and Reference No. 49.
Sincerely,

Reviewer 3 Report
The case-series and review by Barbora Hocková et al. is a very well-designed and written investigation presenting an interesting topic concerning all oral health professionals during the Covid-19 pandemic. The case reports and the review were well described and structured according the scientific guidelines (1, 2). The reviewer recommends only minor changes to improve the quality of the manuscript, these include:
1- The authors described immune dysregulation as one of the pathophysiological pathways of the oral complications but did not discuss this point thoroughly. Please add a more specified discussion of this aspect to explain the specific immunological changes that might occur and which oral complications are related to them.
2- According to the authors' explanation oral complications would rather appear in patients with severe Covid-19 complications due to immunosuppressive and antibiotic treatment besides mechanical ventilation. How would this explain the oral manifestations of Covid-19 in non-hospitalized patients as seen in (3)? Please discuss.
References
(1) Guidelines To Writing A Clinical Case Report. Heart Views. 2017 Jul-Sep;18(3):104-105. doi: 10.4103/1995-705X.217857. PMID: 29184619; PMCID: PMC5686928.
(2) Pautasso M. Ten simple rules for writing a literature review. PLoS Comput Biol. 2013;9(7):e1003149. doi: 10.1371/journal.pcbi.1003149. Epub 2013 Jul 18. PMID: 23874189; PMCID: PMC3715443.
(3) Amorim Dos Santos J, Normando AGC, Carvalho da Silva RL, Acevedo AC, De Luca Canto G, Sugaya N, Santos-Silva AR, Guerra ENS. Oral Manifestations in Patients with COVID-19: A Living Systematic Review. J Dent Res. 2021 Feb;100(2):141-154. doi: 10.1177/0022034520957289. Epub 2020 Sep 11. PMID: 32914677.
Author Response
Dear Reviewer,
On behalf of my fellow co-authors, I would like to thank you for your esteemed effort and time for reviewing our manuscript “Oral Complications of ICU Patients with COVID-19: Case-Series and Review of Two Hundred Ten Cases” submitted to Journal of Clinical Medicine (JCM).
In response to your suggestions, we have amended our manuscript accordingly:
- “The authors described immune dysregulation as one of the pathophysiological pathways of the oral complications but did not discuss this point thoroughly. Please add a more specified discussion of this aspect to explain the specific immunological changes that might occur and which oral complications are related to them.”
We have added a couple of sentences to the discussion part to describe the immune dysregulation as a pathophysiological pathway for the oral lesions, especially in the severely affected COVID-19 patients. Please refer to line no. 327-332.
- “According to the authors' explanation oral complications would rather appear in patients with severe Covid-19 complications due to immunosuppressive and antibiotic treatment besides mechanical ventilation. How would this explain the oral manifestations of Covid-19 in non-hospitalized patients as seen in (3)? Please discuss.”
Actually, there is a number of hypotheses to explain the oral symptoms in COVID-19 patients, they are summarized by the reviews of Riad et al. 2020[1] and Iranmanesh et al. 2020[2]. In our current manuscript, we preferred to focus on the hypotheses which are more relevant to ICU cases whose immune statuses are dramatically affected.
For the non-severe cases, direct viral enanthema,[3] secondary infections,[4],[5] drug reactions[6] had been proposed as pathophysiological pathways. Last summer, our team documented a moderately ill COVID-19 patient who experienced painful white plaque on the tongue corresponding to oral candidiasis, the only explanation for this case was the extended empirical antibiotic use which we depicted as abuse in fact.[7]
Sincerely,
References
- Riad A, Klugar M, Krsek M. COVID‐19 Related Oral Manifestations, Early Disease Features? Oral Dis. Published online June 30, 2020:odi.13516. doi:10.1111/odi.13516
- Iranmanesh B, Amiri R, Zartab H, Aflatoonian M. Oral manifestations of COVID‐19 disease: A review article. Dermatol Ther. Published online November 25, 2020:dth.14578. doi:10.1111/dth.14578
- Jimenez-Cauhe J, Ortega-Quijano D, De Perosanz-Lobo D, et al. Enanthem in Patients with COVID-19 and Skin Rash. JAMA Dermatology. 2020;156(10):1134-1136. doi:10.1001/jamadermatol.2020.2550
- Riad A, Kassem I, Stanek J, Badrah M, Klugarova J, Klugar M. Aphthous Stomatitis in COVID‐19 Patients: Case‐series and Literature Review. Dermatol Ther. Published online January 3, 2021. doi:10.1111/dth.14735
- Riad A, Kassem I, Hockova B, Badrah M, Klugar M. Tongue ulcers associated with SARS‐CoV‐2 infection: A case series. Oral Dis. Published online September 25, 2020:odi.13635. doi:10.1111/odi.13635
- Sakaida T, Isao T, Matsubara A, Nakamura M, Morita A. Unique skin manifestations of COVID-19: Is drug eruption specific to COVID-19? J Dermatol Sci. Published online May 2020. doi:10.1016/j.jdermsci.2020.05.002
- Riad A, Gad A, Hockova B, Klugar M. Oral candidiasis in non‐severe COVID‐19 patients: call for antibiotic stewardship. Oral Surg. Published online October 9, 2020:ors.12561. doi:10.1111/ors.12561

Round 2
Reviewer 1 Report
All time lacks of name of real-time PCR thermocycler. Please write name e.g. CFX96, LightCycler 480, and producer, e.g. Bio-Rad, Eppendorf, Roche of used thermocycler.
Author Response
Dear Reviewer,
We have added the thermocycler and producer. Refer to Line No. 105-107.
Regards,
